# Effects of Footwear on Anterior Cruciate Ligament Forces during Landing in Young Adult Females

**DOI:** 10.3390/life12081119

**Published:** 2022-07-26

**Authors:** Riad Akhundov, Adam L. Bryant, Tim Sayer, Kade Paterson, David J. Saxby, Azadeh Nasseri

**Affiliations:** 1Griffith Centre for Biomedical and Rehabilitation Engineering (GCORE), Menzies Health Institute Queensland, Griffith University, Southport, QLD 4222, Australia; r.akhundov@griffith.edu.au (R.A.); a.nasseri@griffith.edu.au (A.N.); 2Centre for Health, Exercise & Sports Medicine, University of Melbourne, Melbourne, VIC 3010, Australia; albryant@unimelb.edu.au (A.L.B.); sayer.t@unimelb.edu.au (T.S.); kade.paterson@unimelb.edu.au (K.P.)

**Keywords:** low-support, high-support, loading mechanisms, neuromusculoskeletal, biomechanics, ACL injury

## Abstract

Rates of anterior cruciate ligament (ACL) rupture in young people have increased markedly over the past two decades, with females experiencing greater growth in their risk compared to males. In this study, we determined the effects of low- and high-support athletic footwear on ACL loads during a standardized drop–land–lateral jump in 23 late-/post-pubertal females. Each participant performed the task unshod, wearing low- (Zaraca, ASICS) or high- (Kayano, ASICS) support shoes (in random order), and three-dimensional body motions, ground-reaction forces, and surface electromyograms were synchronously acquired. These data were then used in a validated computational model of ACL loading. One-dimensional statistical parametric mapping paired *t*-tests were used to compare ACL loads between footwear conditions during the stance phase of the task. Participants generated lower ACL forces during push-off when shod (Kayano: 624 N at 71–84% of stance; Zaraca: 616 N at 68–86% of stance) compared to barefoot (770 N and 740 N, respectively). No significant differences in ACL force were observed between the task performed wearing low- compared to high-support shoes. Compared to barefoot, both shoe types significantly lowered push-off phase peak ACL forces, potentially lowering risk of ACL injury during performance of similar tasks in sport and recreation.

## 1. Introduction

Rates of anterior cruciate ligament (ACL) rupture have increased precipitously over the past several decades in developed nations [1]. This increase has occurred despite awareness of the serious long-term consequences for knee health following ACL injury [2] and efforts to reduce injury risk. Pubertal and young adults have experienced particularly large increases in rates of ACL injury [3], with young females at elevated risk for ACL rupture compared to their male counterparts [1,4] even when accounting for their lower relative exposure to athletic activity [5]. The natural questions follow, why do females at, or around, the time of sexual maturation rupture their ACL so frequently, and what can be done to ameliorate the situation?

Anatomical (e.g., lower-limb alignment [6], knee anatomy [7]), neuromuscular (e.g., body posture [8,9], muscle strength [8], recruitment patterns [10]), and incidental (e.g., play intensity [5], sport type [11]) factors have each shown forms of statistical association with ACL injury established through retrospective or prospective study. General exercise-based injury prevention programs are cost-effective [12], and those aimed at reducing ACL injury in young females have been shown to be moderately protective [13]. Although exercise-based programs have been shown to reduce ACL injury rates (among the myriad other health benefits they provide), exercise as an ACL injury prophylactic is associated with numerous barriers (e.g., compliance, fidelity, etc.) that may limit success at scale (e.g., national levels). In contrast, a passive external tool that lowers ACL forces during provocative athletic tasks might be a viable method to affect ACL injury rates at scale.

Preliminary evidence suggests footwear might affect ACL loading. In the current market, a wide variety of low- and high-support athletic footwear options exist. Although no international consensus exists as to the definition of “support” in footwear, the recommendations from the Footwear Assessment Tool [14] have been used previously [15] to identify footwear features that may alter external knee loading. Therein, high-support footwear possesses a medial post (i.e., midsole that is stiffer medially compared to laterally), increased longitudinal bending stiffness, and midfoot rotational stability to minimize excessive foot pronation during activity, whereas low-support footwear does not [16]. Although motion-control shoes have been shown to reduce risk of pronation-related injuries [17], no footwear features have been associated with altered risk of ACL injury. However, modifying medial and lateral support in footwear does affect frontal and transverse-plane knee loads [18,19,20], which may affect ACL loads [21]. Furthermore, both low- and high-support shoes have been shown to increase external knee flexion moments compared to barefoot in females during a standardized drop–land–jump task [15]. Collectively, footwear has been shown to alter external knee loads, which might affect ACL loading, although this remains to be investigated.

Importantly, as has been demonstrated numerous times, external knee biomechanics (e.g., knee flexion moment) are poor indicators of load sharing amongst the many internal structures of the knee (e.g., ligaments and articular surfaces) [22,23,24]. Changes in posture, external loading, and muscle activation patterns (inferred through electromyography) can have complex and non-intuitive effects on load sharing within the knee [25]. To gain clarity, we developed and validated a computational model that accurately estimates ACL forces during dynamics tasks [26,27]. In this study, we determined whether and how low- and high-support shoes modulate ACL force during a standardized drop–land–jump task in late-/post-pubertal females. As previous analysis of females performing a drop–land–jump task revealed, both low- and high-support shoes resulted in larger external knee flexion moments compared to barefoot performance [15]. Hence, we hypothesized that compared to barefoot, ACL forces during the drop–land–jump would be greater when wearing either low- or high-support shoes.

## 2. Materials and Methods

Twenty-three healthy late-/post-pubertal females (i.e., Tanner stages IV and V with growth spurt and menarche [28,29]), were recruited from local universities, schools, sporting facilities, and community centres. Participants were informed of the study via advertisements in newsletters, posters, and/or brochures. All interested participants (and/or parents/guardians) were initially contacted via telephone or email to address any potential concerns/questions regarding the study procedure. Once satisfied, participants > 18 years of age and parents/guardians of daughters < 18 years of age were sent an online screening form via e-mail. The screening form determined (i.e., yes/no response) whether each of the study’s inclusion/exclusion criteria were satisfied. Inclusion criteria were recreationally active participants (i.e., ≥30 min of moderate or vigorous daily physical activity confirmed via a self-reported physical activity questionnaire [30]) with a body mass index < 30 kg·m^−2^. Exclusion criteria were history of lower limb injury, knee pain, medical condition affecting performance of sporting tasks, or previous ACL or meniscal injuries. Prior to data collection, approval was obtained from the University of Melbourne Human Research Ethics Committee (no. 1442604), and all participants or parents/guardians of those < 18 years of age provided written informed consent.

Data acquisition was conducted at the Centre for Health, Exercise & Sports Medicine, University of Melbourne, Australia. After task familiarization, participants performed three trials of a standardized drop–land–lateral jump for each footwear condition: barefoot, low-support shoes (Zaraca, ASICS), and high-support shoes (Kayano, ASICS). We defined low- and high-support footwear consistent with the categorization in [15]. Briefly, “high-support” was defined to include midsoles with higher density/stiffness medially compared to laterally, <10° torsional stiffness, <10° heel counter stiffness, and <45° midfoot longitudinal stability. Low-support was defined by uniform midsole density, 10–45° heel counter stiffness, 10–45° torsional stiffness, and >45° midfoot longitudinal stiffness. At the time of the original study [15] from which data for the current study were obtained the only neutral ASICS shoe with similar cushioning features to ASICS Kayano but classified “low-support” based on our criteria was ASICS Zaraca. We acknowledge that the cushioning is slightly different between the two shoes, as both contain ASICS rearfoot and forefoot gel, whereas Zaraca has SpEVA and Kayano ‘Flytefoam’. However, there is no evidence that these different materials result in differential effects on external knee loading, muscle activations, ACL force, or any other kinematic or kinetic variable.

The performance order of footwear and barefoot was randomized to avoid order effects. The drop–land–lateral jump, represented by silhouettes in Figure 1, began with participants standing on their dominant leg at the centre of a box with their hands folded across their chest. Box height was normalized to 30% of each participant’s leg length, and the box was positioned 10 cm from a ground-embedded force platform (2400 Hz, AMTI, Watertown, MA, USA). Participants then dropped down, landed on the force platform with their dominant leg, and immediately performed a 90° lateral jump, landing on their contralateral leg atop a marked target at 150% of their leg length from the centre of the force platform. This task was selected based on its similarity to single-leg landing manoeuvres performed in many sports, given that the majority of ACL ruptures occur during single-leg stance [9].

During the drop–land–lateral jump, three-dimensional ground reaction forces (GRF) were acquired using the ground-embedded force platform. Whole-body motion was acquired using a 12-camera motion capture system (Vicon Motion Systems, Oxford, UK), which sampled at 120 Hz, as well as a whole-body retroreflective marker set adhered to the skin atop specific anatomic landmarks [31]. Muscle activations were measured using a wireless surface EMG system (Noraxon, Scottsdale, AZ, USA) sampling at 2400 Hz. Surface EMG electrodes were fixed atop the eight major lower limb muscles spanning the knee of the dominant leg (i.e., rectus femoris, vastus lateralis, vastus medialis, tibialis anterior, lateral gastrocnemius, medial gastrocnemius, lateral hamstrings, and medial hamstrings), consistent with the Surface Electromyography for the Noninvasive Assessment of Muscles guidelines (http://www.seniam.org/). Using the MoToNMS [32] toolbox for MATLAB (version 2019a, Mathworks, Natick, MA, USA), marker trajectories and GRF were low-pass-filtered with a zero-lag, 4th-order 6 Hz Butterworth filter. The EMG signals were first band-pass-filtered (30–300 Hz), full-wave-rectified, and then low-pass-filtered using a cut-off frequency of 6 Hz to produce linear envelopes. The EMG signals and their corresponding linear envelopes were input into an EMG classification tool [33] to assess EMG quality and appropriateness for neuromusculoskeletal modelling. Each resulting EMG linear envelope was subsequently amplitude-normalized to its maximum envelope value identified from all available trials [34].

A modified generic, full-body model [24,35] was used to perform musculoskeletal modelling in OpenSim [36] version 3.3 (Stanford University, Palo Alto, CA, USA). The model consisted of 37 degrees of freedom and 80 muscle–tendon unit actuators. The model was linearly scaled using bony landmarks and hip-joint centres to match each participant’s body segment dimensions, mass, and inertia. Optimal muscle fibre and tendon slack lengths were morphometrically scaled to preserve physiological operating ranges for muscle and tendon [37]. Each muscle’s maximum isometric strength was then updated using the mass–height–muscle volume relationship [38]. After model preparation, model-generalized coordinates (i.e., motions), generalized loads (i.e., net joint forces and moments), and muscle–tendon unit (MTU) kinematics (i.e., moment arms and lines of action) were computed in OpenSim for each trial using inverse kinematics, inverse dynamics, and muscle analysis tools, respectively [36,39].

Normalized EMG linear envelopes, generalized loads, and MTU kinematics were subsequently used in the Calibrated EMG-informed Neuromusculoskeletal Modelling Toolbox [32] (CEINMS) to estimate lower-limb muscle dynamics. First, CEINMS was used to synthetize excitations for muscles with no experimental EMG or with EMG quality classified as unsuitable for direct use in neuromusculoskeletal modelling by the EMG classification tool. Then, CEINMS calibration adjusted initial values of tendon slack length, optimal fibre length, pennation angle, activation dynamics parameters, and maximum isometric force of each muscle of the participant’s model in a subject-specific manner. The objective of the calibration was to minimize torque tracking error at the knee (i.e., squared error between inverse dynamics and CEINMS-generated knee moments). After each participant’s model was calibrated, CEINMS was used in EMG-assisted mode to estimate lower-limb muscle and tibiofemoral contact forces, fibre lengths, and fibre velocities. Finally, ACL force and its uniplanar (sagittal, transverse, and frontal) causal contributors throughout the stance phase of the drop–land–lateral jump were determined by incorporating lower-limb biomechanics from OpenSim and CEINMS into a validated ACL force model [26,27].

Stance was defined as the period between the first and last instances of foot-to-ground contact of the instrumented leg, as detected by the vertical GRF exceeding 20 N. For each trial, ACL forces were normalized to 100% of stance, averaged for each participant, and ensemble-averaged for each footwear condition. Statistical parametric mapping (SPM) was used to analyse ACL force in a time-continuous manner [40]. The SPM analyses were implemented using open-source SPM1d software (v.M.0.4.7, www.spm1d.org) (accessed on 20 April 2022) in MATLAB version 2019a. Data normality was verified using SPM1d, and subsequently, one-dimensional SPM paired t-tests (α = 0.05) were used to compare ACL force during stance and its contributors between the footwear conditions.

## 3. Results

During the drop–land–lateral jump, participants (age = 19.7 ± 4.0 years, height = 1.65 ± 0.06 m, body mass = 59.7 ± 9.5 kg, body mass index = 21.9 ± 3.5 kg·m^−2^) generated significantly lower total ACL forces during the push-off phase while wearing low- (average of 616 N at 68–86% of stance) and high- (average of 624 N at 71–84% of stance) support shoes compared with barefoot (average of 770 N and 740 N, respectively). The timing of peak ACL forces remained unchanged by the presence of footwear. The ACL force developed through the sagittal plane was significantly lower when participants wore low- (average of 583 N at 67–86% of stance) and high- (average of 584 N at 71–85% of stance) support shoes compared to barefoot (731 N and 706 N, on average, respectively). The ACL force developed through the transverse plane was significantly higher while wearing high-support shoes (average of 41 N at 0–5% of stance and 81 N at 12–20% of stance) compared with barefoot (31 N and 70 N on average, respectively). Additionally, low-support shoes showed significantly higher (average of 59 N at 0–18% of stance) ACL force around foot-strike and significantly lower (average of 54 N at 58–68% of stance) ACL force at push-off compared with barefoot (51 N and 61 N on average, respectively). No significant differences were observed in the frontal plane contributions to ACL force between the three footwear conditions. Likewise, total ACL force and its uniplanar contributions showed no differences when compared between the low- and high-support shoes.

## 4. Discussion

The current study determined effects of low- and high-support footwear on ACL forces during a provocative motor task in late-/post-pubertal females. Furthermore, uniplanar contributions to ACL force were explored. Using a neuromusculoskeletal modelling approach, we determined ACL forces generated during a drop–land–jump were reduced when participants wore either low- or high-support shoes compared with the ACL forces generated performing the same task barefoot. We did not find any significant differences between the ACL force generated wearing low- and high-support shoes. As this study used legacy data [41], it was not intentionally powered to discriminate effects on ACL force between the two tested footwear models. Validated ACL force models did not exist at the time of study inception. Consequently, it was unsurprising that we did not find evidence of footwear-specific effects on ACL forces. Due to the recent advent of a validated ACL force model [26], footwear designs to date have not been informed by modelling of ACL loading during athletic tasks. Future footwear designs could potentially involve a “human-in-loop paradigm” to identify design features that reliably reduce ACL forces during provocative athletic tasks.

Compared to barefoot, wearing either low- or high-support shoes reduced ACL forces during the drop–land–jump. This result ran contrary to our hypothesis, which was informed by prior analysis of a similar cohort of participants [15], which reported increased external knee flexion moments when wearing low- and high-support shoes compared with barefoot performance. We are again reminded that external biomechanics (e.g., knee flexion moments) are poor indicators of internal load sharing (e.g., ACL force). Although both footwear models tested in the present study significantly lowered ACL force relative to barefoot, the reduction did not occur during landing but during push off. Analysis of ACL rupture incidents captured on video indicates rupture occurs shortly following initial ground contact (i.e., 30–100 ms), which is consilient with measurements of sophisticated robotic simulation of ruptures using cadavers [42]. Therefore, footwear-associated reductions in ACL force observed in this study may have little relevance to reduction in risk of ACL injury.

We found no statistically significant differences in ACL forces generated wearing low- compared to high-support shoes. This finding is understandable, as the design features present in the low- and high-support shoes tested in this study were not developed specifically to affect ACL forces. Indeed, no validated ACL models were available to manufacturers at the time of product development. Moreover, the sensorimotor effects of wearing any type of athletic footwear (be it low- or high-support) are likely much more potent than the variations in sensorimotor experience created by the different footwear models. Numerous studies [43] have shown that footwear features, such as textured or vibrating insoles, affect mechanical and sensorimotor responses, resulting in improved gait patterns. However, these studies used orthoses with more prominent design features than those present in the Kayano and Zaraka models tested in this study. Furthermore, those studies were conducted primarily on elderly people walking, not young adults performing a vigorous motor task, such as a drop–land–jump. The design features in the low- and high-support shoes tested in the present study appear insufficient to elicit differential responses in ACL force and are thus unlikely to impact injury risk. However, a validated ACL force model now exists [26], so it may be possible to develop footwear features capable of reliably reducing ACL forces during relevant phases of motor tasks known to be associated with ACL rupture (e.g., side-step cutting, single-leg landing), implying that if ACL loads are lowered and if all else remains constant, ACL injury will be less likely.

Some limitations of this study should be considered. First, the reductions in ACL force due to footwear were immediate effects. It is unclear whether lower ACL forces would be sustained over time or whether greater familiarity with the footwear would result in ACL forces returning to values similar to those generated barefoot. Second, we examined only low- and high-support shoe variations supplied by the original study’s [41] industry partner, ASICS Oceania. We cannot comment on whether the effects observed in this study are peculiar to the tested Kayano and Zaraca models or are generalizable to other makes and models. Third, as we aimed to make a specific comparison between low- versus high-support shoes, we selected Kayano and Zaraca models, which are running shoes. We acknowledge other shoe types are often used for sports that have high rates of ACL injury, such as netball and handball; however, we could not make a shoe-support-focused comparison for any footwear type other than running. Fourth, the drop–land–lateral jump task was standardized to limit biomechanical variability when participants performed multiple repeats of the same task and to facilitate comparison between cohorts of differing developmental maturity [44,45]. A consequence of this standardization might be that the task was unlikely to seriously challenge the neuromuscular system or threaten knee injury. It would be highly unethical to deliberately subject a participant to such risk. This means the standardized drop–land–lateral jump did not simulate real-world scenarios where the knee experiences substantial risk of ACL injury (e.g., unanticipated side-step cut under pressure from defenders). Finally, this was a secondary analysis of a subsample from a legacy dataset; thus, it was not designed to detect differences between the effects of footwear on ACL force during a landing task but did generate important data from which effect sizes can be estimated to power future studies.

Despite these limitations, this study is the first to compare effects of footwear on ACL force. The practical implications of this study are primarily relevant to footwear companies. The fact that no differential effects on ACL loading were found between low- and high-support shoes suggests the designs we tested were not capable of modulating ACL force, the design mix contained antagonistic features (e.g., one feature would increase, whereas another would decrease ACL force, nullifying overall effect), and/or responses to footwear were specific to the individual (i.e., for some, model X increases ACL force relative to model Y and vice versa or equivalence for other individuals). The first two explanations of our null result could be addressed in silico, whereby design features are simulated in isolation and in combination (i.e., Monte Carlo) and then with an objective (i.e., optimization to lower ACL force while preserving performance). Simulated results could then be tested experimentally to confirm or refute effects. A non-trivial challenge of in silico design and simulation is to robustly account for the neural effects of design modifications (i.e., changing muscle activation patterns in response to different footwear). Despite these acknowledged challenges, we contend that simulation is superior to the current “cut and try” approach, as it enables companies to test more variants than could ever be evaluated empirically, reduces costs associated with physical experimentation by eliminating unlikely solutions from the test pool, and potentially reveals non-intuitive design combinations through computational optimization.

In conclusion, ACL forces generated during performance of a drop–land–lateral jump task were of lower magnitude when participants wore either low- or high-support shoes compared with barefoot. Although no differences in ACL force were found between the two footwear models tested, results indicate that footwear has the potential to reduce ACL loading when performing standardized motor tasks compared to barefoot. In the future, footwear manufactures could use the computational models presented herein to inform their designs and evaluate the effectiveness of footwear to lower ACL force.

## Figures and Tables

**Figure 1 life-12-01119-f001:**
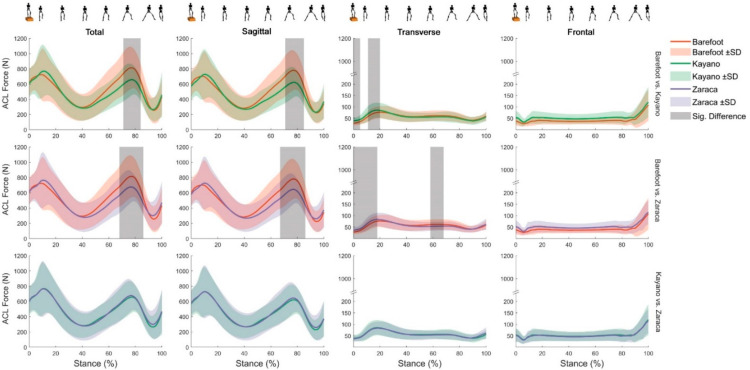
Force (N) in the ACL during drop–land–lateral jump. Barefoot (red), Kayano (green), Zaraca (purple), and their respective variations across participants presented as standard deviation (shaded areas). Areas of significant difference, as determined by statistical parametric mapping (grey area). Silhouettes atop the figures represents the progression through drop–land–lateral jump.

## Data Availability

The data supporting the findings presented in this manuscript are available from the corresponding author upon reasonable request.

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
