# Peer review of "Effects of Footwear on Anterior Cruciate Ligament Forces during Landing in Young Adult Females"

_life, 2022, doi:10.3390/life12081119_

Round 1

Reviewer 1 Report

The authors would benefit from the book

Biomechanics

M Doblare, J Merodio EOLSS Publications   and several of their chapters that deal with aspects of this (interesting) paper, This reference could be included since it is an state of the art.

Author Response

Please see attached document. Thank you kindly.

Reviewer 2 Report

The aim of the study was to assess the effects of low- and high-support shoes on anterior cruciate ligament (ACL) forces during a drop-land-lateral jump in late-/post-pubertal females. The forces registered in the ACL during jumps, specifically during the push off phase, were significantly lower when wearing both shoe models compared to barefoot, with no differences between shoe conditions. The authors concluded that footwear has potential to reduce ACL loading and therefore, injury risk.

In the next paragraphs, I make a couple of comments and suggestions intended to help authors to improve the quality of their manuscript.

Introduction:

In my opinion, the information provided concerning sport footwear do not established a robust background. I suggest authors to add more scientific evidence regarding the effects of sport footwear and its characteristics (e.g., cushioning materials, midsoles, drop, etc.) on lower limbs biomechanics, since this is a key point of your investigation.

Materials and methods:

Why did you use these two different types of running shoes? The Asics Gel-Kayano are running shoes for overpronation, with support on the foot arch, while the Asics Gel-Zaraca are neutral running shoes (i.e., without support). Apart from this, the cushioning characteristics of both models are quite different. The Gel-Kayano is a top-model of the brand Asics and provides one of the best cushioning systems offered by this brand. The neutral model comparable to the Gel-Kayano, regarding cushioning and materials characteristics, would be the Asics Gel-Nimbus, while Gel-Zaraca is considered a “mid-range” model. These aspects can affect lower limbs biomechanics when performing different exercises, especially when these tasks imply landing (e.g., jumps). I suggest to discuss this issue in the discussion section. I also recommend authors to briefly describe the footwear characteristics and features in the materials and methods section.

Discussion:

Lines 192-195: In relation to my precedent comment, why did you use running shoes and not another type of sport footwear (e.g., basketball, handball) related to those sports modalities with a higher prevalence of ACL injury?

Please include some practical applications derived from your findings.

Author Response

Please see the attached document. Thank you kindly.

Reviewer 3 Report

The manuscript is well-written. I provide some minor commentaries:

Introduction

There is a need to state the differences between both (high-low) type of shoes and also whether they are related to ACL injury prevalence.

Provide a more detailed rationale for the hypothesis stablished.

Methods

It is not clear how participants were recruited (email, personal invitation..) and from where (sport centre, sport team…).

All of the participants initially recruited were already active or some of them had to be excluded from the research?

Did the participants have experience in sports competition (individual/team) or they were just active people who performed recreational physical activity?

After revising the funding sources I imagine that ASICS has nothing to do with the research. If so, why did the authors chose this brand and no other? Any particular reason?

Results

Please add the characteristics of the sample

Discusion

A further limitation is the fact that the sample was made up of active people. It is mandatory to clarify their characteristics (recreational/competitive athletes) for a better understanding of these limitations.

Please, elaborate on this idea “results indicated footwear has potential to reduce ACL loading when 228 performing athletic tasks”. Initially, it seems that it does not matter whether athletes are barefoot or wear high/short shoes. This relates to my previous suggestion indicating the impact that the type of shoe had on ACL injury prevalence.

Author Response

(The authors gave the same response as above.)

Round 2

Reviewer 3 Report

My suggestions have been properly addressed. I have no further comments.